# Aerobic-Resistance Training with Royal Jelly Supplementation Has a Synergistic Effect on Paraoxonase 1 Changes and Liver Function in Women with MASLD

**DOI:** 10.3390/medicina61020349

**Published:** 2025-02-17

**Authors:** Roya Askari, Nazanin Rabani, Hamid Marefati, Marzie Sadat Azarnive, Matteo Pusceddu, Gian Mario Migliaccio

**Affiliations:** 1Department of Exercise Physiology, Faculty of Sports Sciences, Hakim Sabzevari University, Sabzevar 9617976487, Iran; h.marefati@hsu.ac.ir; 2Department of Sports Sciences, Faculty of Sports Sciences, Hakim Sabzevari University, Sabzevar 9617976487, Iran; nazanin.rb56@gmail.com; 3Department of Sports Sciences, Faculty of Literature and Humanities, Zabol University, Zabol 9861615881, Iran; m.azarnive@uoz.ac.ir; 4Department of Human Sciences and Promotion of Quality of Life, San Raffaele Open University, 00166 Rome, Italy; pusceddumatteo@tiscali.it; 5Athlete Physiology, Psychology and Nutrition Unit, Maxima Performa, 20126 Milan, Italy

**Keywords:** aerobic-resistance training, royal jelly, paraoxonase 1, oxidized LDL, liver function, lipid profile, postmenopausal women, MASLD

## Abstract

*Background and Objectives*: Metabolic dysfunction-associated steatotic liver disease (MASLD) is a clinical pathological syndrome characterized by steatosis and fat accumulation in liver parenchymal cells in patients without a history of excessive alcohol drinking. Currently, there is no definitive treatment for MASLD, and its prevalence increases with age and obesity, and after menopause. Among the ways to treat it, we can mention regular sports exercises and the use of natural supplements. Therefore, the aim of this research is to investigate and compare the effects of aerobic-resistance training with royal jelly supplementation on changes in paraoxonase 1, oxidized LDL, liver function, and lipid profile in postmenopausal women with Dysfunction-Associated Steatotic Liver Disease. *Materials and Methods*: This semi-experimental study involved 23 women with Dysfunction-Associated Steatotic Liver Disease with an average weight (71.34 ± 11.63 kg), age (48.54 ± 3.88 years), and body mass index (27.63 ± 4.20 kg/m^2^). They were randomly divided into two groups: exercise + supplement (*n* = 12) and exercise + placebo (*n* = 11). Both groups performed eight-station resistance exercises (8–12 repetitions in 2–4 sets) for 8 weeks, with three sessions per week (for 35–40 min, from 10-15 RPE), and then, for 10–15 min of active rest, they performed aerobic exercises with an intensity of 40–85% of the target heart rate, in two-minute intervals with 45 s of active rest. Royal jelly supplement (500 mg on training days, before each training session) was consumed. Blood sampling was done before and 48 h after the last training session. Statistical analysis was performed using a variance test with repeated measures (two groups × two stages of pre-test-post-test) in SPSS software (Version 26) with a significance level of *p* < 0.05. *Results*: The results of the statistical analysis show that the effects of eight weeks of exercise + supplement and exercise + placebo on PON1, oxLDL, lipid profiles (HDL, LDL, TC, and TG), and liver enzymes (ALT, AST) in women with non-alcoholic fatty liver showed a significant difference (*p* < 0.05). The results show a significant increase in PON1 (*p* = 0.008) and HDL (*p* = 0.005) in the exercise + supplement group compared to the exercise + placebo group. But significant decreases in oxLDL (*p* = 0.031), TC (*p* = 0.045), TG (*p* = 0.013), LDL (*p* = 0.027), ALT (*p* = 0.015) and AST (*p* = 0.009) were observed in the exercise + supplement group compared to the exercise + placebo group (<0.05). The results show a significant increase in PON1 (*p* = 0.008) and HDL (*p* = 0.005) in the exercise + supplement group compared to the exercise + placebo group. However, significant decreases in oxLDL (*p* = 0.031), TC (*p* = 0.045), TG (*p* = 0.013), LDL (*p* = 0.027), ALT (*p* = 0.015), and AST (*p* = 0.009) was observed in the exercise + supplement group compared to the exercise + placebo group. *Conclusions*: Based on the results, it can be concluded that aerobic-resistance exercises with the addition of royal jelly can probably be an efficient and recommended strategy to minimize the harmful effects of Dysfunction-Associated Steatotic Liver Disease by affecting the activity of liver enzymes, paraoxonase 1, LDL oxidation, and lipid profile. Although exercise alone also yielded favorable results, according to the findings of this research, it can be said that exercise, combined with the use of royal jelly supplements, may have more positive effects on reducing liver complications and improving body function. However, in order to obtain more accurate scientific evidence, it is necessary to investigate more doses and timing of royal jelly in future studies.

## 1. Introduction

Metabolic Dysfunction-Associated Steatotic Liver Disease (MASLD) is a clinical pathological syndrome characterized by steatosis and the accumulation of fat within the parenchymal cells of the liver, occurring in the absence of a history of excessive alcohol consumption in the affected individual [1]. Currently, there is no definitive treatment available for MASLD [2]. The prevalence of this condition increases with advancing age, obesity, and post-menopausal status, and it is reported to be twice as common in post-menopausal women [3]. During menopause, a deficiency of estrogen contributes to the accumulation of visceral fat, elevated triglyceride and cholesterol levels, and a reduction in energy expenditure, ultimately leading to enhanced lipogenesis, fat accumulation in the liver, and the onset of inflammation [4].

Increased oxidative stress and inflammation are pivotal contributors to the onset and progression of liver disease. One important biomarker indicative of the severity of liver dysfunction or Dysfunction-Associated Steatotic Liver Disease is paraoxonase, specifically the activity of the enzyme paraoxonase-1 (PON1) [5]. The PON1 enzyme functions as an antioxidant, mitigating oxidative stress through the hydrolysis of lipoperoxides. Given the elevation of beta-oxidation of fatty acids, alongside the production of lipid peroxides and oxidative stress associated with Dysfunction-Associated Steatotic Liver Disease, the significance of the PON1 enzyme becomes evident [6,7].

Given that both lipid metabolism and oxidative stress are significant components of the pathogenesis of non-alcoholic steatohepatitis (NASH), it is not surprising that these processes are interconnected and result in the production of oxidized inflammatory lipids that adversely affect liver health. One of the most extensively studied examples of an oxidized lipoprotein associated with inflammation is oxidized low-density lipoprotein (oxLDL). OxLDL is widely regarded as a toxic compound capable of eliciting detrimental inflammatory responses in the liver and other related organs [8]. On the one hand, the utilization of supplements and medications, and adherence to a healthy lifestyle, can serve to prevent the progression of diseases and promote individual improvement. Recently, due to the adverse side effects associated with conventional medications, a significant number of researchers and health specialists have shifted their focus toward the application of natural supplements [9], among which royal jelly is considered one of the most significant. Royal jelly is a viscous, lemon-colored substance that is water-soluble and exhibits acidic properties. It is secreted by the hypopharyngeal and mandibular glands of young worker honeybees, and is consumed by the larvae within the hive for a limited duration and by the queen bee throughout her lifespan [10].

Exercise is not only one of the foundations of a healthy lifestyle, but also a critical factor influencing PON1 activity [11]. Research has demonstrated that aerobic exercise reduces the expression of pro-inflammatory markers in the liver and promotes a shift in Kupffer cells towards a more anti-inflammatory phenotype via metabolic reprogramming [12,13]. Resistance training can also induce beneficial changes in the health of visceral adipose tissue (VAT) by remodeling the extracellular matrix, decreasing the size of adipocytes, lowering inflammatory markers, and increasing muscle mass [14]. Notably, a 13% reduction in liver fat was reported in participants with MASLD and Type 2 diabetes after eight weeks of resistance training, occurring without any concomitant weight loss [15,16].

However, the existing literature indicates that studies examining combined exercise interventions in individuals with MASLD remain limited [17]. Resistance training [18] has yielded sometimes contradictory results [18,19]. In a study conducted by Banitalebi et al. (2018) [20], 10 weeks of combined training resulted in a significant reduction in visceral fat and liver fat accumulation, and improvements in the patients’ condition. Therefore, combined exercise is recommended for enhancing fatty liver conditions in women.

The utilization of natural products, particularly royal jelly, has recently garnered significant attention in the academic community [21,22]. Numerous clinical studies have investigated the antioxidant properties, improvements in lipid profiles, and anti-inflammatory effects associated with royal jelly [23]. However, despite the extensive body of research, no studies have concurrently examined the effects of combined exercise and royal jelly supplementation on these parameters in patients diagnosed with MASLD. Therefore, the objective of this research was to investigate the effects of an 8-week regimen of aerobic-resistance training, with and without royal jelly supplementation, on alterations in paraoxonase 1, liver function, oxidized low-density lipoprotein (OxLDL), and lipid profiles in women suffering from Dysfunction-Associated Steatotic Liver Disease.

## 2. Materials and Methods

The current study employed a semi-experimental applied design utilizing a pre-test–post-test framework. The target population comprised women in Mashhad diagnosed with stage one Dysfunction-Associated Steatotic Liver Disease. Participants were recruited through a public announcement and subsequently confirmed by a medical professional. A total of 30 individuals were selected based on specific inclusion criteria, which included general health status, the ability to engage in physical exercise, an age range of 45 to 55 years, a diagnosis of stage one Dysfunction-Associated Steatotic Liver Disease, a minimum of one year post-menopause, and the absence of any dietary or exercise regimen for the preceding six months. Their fatty liver was confirmed through ultrasound, with subsequent diagnosis and interpretation provided by a specialist in endocrinology and metabolism.

In the initial and final weeks of the intervention, participants completed a 24 h dietary recall questionnaire. Throughout the duration of the intervention, no modifications were made to the participants’ dietary regimens. Ultimately, the average intake of energy, fiber, sugar, protein, carbohydrates, and fat for both groups was analyzed using N4 software (version 3.5.2) revealing no significant differences in these values.

Initially, based on previous studies, the sample size was determined using G*Power software (version 3.1.9.7)which considered an effect size of 0.3, an alpha level of 0.05, and a power of 0.95. This analysis suggested a sample size of 24 participants; however, this number was subsequently increased to 30 to account for potential attrition.

Participants were randomly assigned to two groups: (1) a combined exercise group receiving a placebo and (2) a combined exercise group receiving royal jelly supplementation. During the exercise program, seven participants were excluded due to unforeseen circumstances, including illness, personal issues, or irregular attendance at training sessions. Ultimately, data were collected and analyzed for 23 individuals (12 in the supplementation group and 11 in the placebo group). This study was conducted with ethics approval under identification code IR.HSU.REC.1402.024 from Hakim Sabzevari University. To monitor the dietary intake of the participants, a 24 h dietary recall questionnaire was administered at baseline, one week prior to the commencement of the study, and during the final week of the intervention. Following participant selection, comprehensive information regarding the exercise programs and evaluations was provided during an orientation session, and participants completed a consent form as well as the relevant questionnaires. A blood sample was collected one week prior to the initiation of the first training session and again during the post-test phase. The combined exercise program spanned eight weeks, with three sessions conducted each week. Additionally, 500 mg capsules of a uniform color were utilized for both the supplementation and placebo groups. The supplementation was administered in a double-blind manner, with 500 mg of royal jelly contained within the oral capsules, while similar capsules filled with starch powder were designated for the placebo group. Both the supplement and placebo were consumed by the participants one hour prior to each training session.

### 2.1. Exercise Protocol

The sequence of exercise in this study involved an initial phase of resistance exercises utilizing elastic bands, followed by aerobic exercises conducted on a treadmill. Each exercise session commenced with a general warm-up lasting 10 min, which included 5 min of walking and 5 min of specific dynamic mobility exercises. This was succeeded by a resistance training regimen employing elastic bands for a duration of 35 to 40 min, structured in a circuit format. Following a 10 to 15 min active rest period, participants engaged in aerobic exercises on a treadmill, concluding with a 5 min cool-down that incorporated stretching movements for both upper and lower limbs.

The resistance training program targeted both upper and lower muscle groups and consisted of exercises such as seated elbow flexion, standing elbow extension, upper body extensors (chest and triceps), upper body flexors (latissimus dorsi and trapezius), calf raises while lying down, knee flexion in a lying position, hip abduction and adduction while standing, and seated ankle flexion and extension. The temporal parameters for each movement were standardized to “2 s eccentric, 1 s isometric, and 1 s concentric”. The load and intensity of the exercises were progressively increased throughout the intervention. Intensity for each stage was determined based on the color of the bands employed and the Rate of Perceived Exertion (RPE) scale, which ranged between 10 and 15, as well as additional sets and changes in band color. This progressive overload approach ensured that participants were sufficiently challenged while maintaining safety within their workout regimen.

Table 1 delineates the specific resistance levels associated with various band colors and their corresponding RPE scores. Prior to the commencement of the exercise program, participants underwent one week of familiarization with the correct execution of movements and the RPE scale. The initial resistance load for all participants was standardized using yellow elastic bands. Every two weeks, the training program was reassessed, and band color was modified according to individual variations in performance and progress. During the aerobic phase, exercises were performed in 2 min intervals, interspersed with 45 s of active rest, until participants achieved 40% of their target heart rate on the treadmill.

Over the eight-week duration, participants attained 70–85% of their predicted maximum target heart rate as calculated using the Karvonen formula. In addition to monitoring the increase in intensity through heart rate assessments, the number of repetitions progressed from 8 to 12, tailored to each participant’s ability and fitness level throughout the intervention. The additional load was adjusted according to the participants’ tolerance and progress, ensuring that the perceived exertion score, measured using the Borg Rating of Perceived Exertion (RPE) scale, was maintained within the range of 10 to 15 (as illustrated in Table 2). This systematic approach facilitated appropriate and safe progression for all participants involved in the study.

### 2.2. Blood Sampling

Blood samples were collected between 7:00 and 9:00 a.m. following a 12 h fasting period, one week prior to the initial exercise session and 48 h after the final training session, in order to mitigate any potential short-term effects of exercise. Pre-test standardization protocols included the absence of exercise, adherence to standardized meals, and a minimum of 8 h of sleep. A total volume of 10 mL of blood was drawn from the antecubital vein. The samples were collected in tubes containing an anticoagulant with a concentration of 3 to 4 mg/mL ethylene diamine tetraacetic acid. They were promptly centrifuged at 2000× *g* rpm for 10 min, and the resulting plasma was stored in a freezer at −80 °C until the time of analysis.

The plasma was subsequently prepared for the analysis of total cholesterol (TC), high-density lipoprotein (HDL), oxidized low-density lipoprotein (oxLDL), low-density lipoprotein (LDL), triglycerides (TG), Paraoxonase 1 (PON1), and liver enzymes (ALT and AST). The levels of TC, HDL, oxLDL, LDL, and TG were quantified using an automatic chemistry analyzer in conjunction with enzymatic assay kits. The concentrations of PON1 and liver enzymes were determined using an enzyme-linked immunosorbent assay (ELISA) methods with specific human kits sourced from (ZellBi Hinter den Gärten, 5689173 Lonsee, Germany), which exhibit a sensitivity range of 5 to 150 nmol/L. This standardized methodology ensured accurate measurement and reliable results for the biochemical markers of interest in the study.

### 2.3. Statistical Analysis

Data were analyzed utilizing both descriptive and inferential statistics. To assess the normality of the data distribution, the Shapiro–Wilk test was employed. Variance homogeneity was evaluated using Levene’s test. In the inferential statistics section, a repeated measures ANOVA (two groups × two stages of pre-test and post-test) along with Bonferroni’s post hoc test were conducted to analyze the data. All calculations were executed using SPSS software version 26, with a significance level set at 0.05 (*p* < 0.05) and a 95% confidence interval.

## 3. Result

Demographic and physiological information of the participants by group is presented in Table 3.

The outcomes of the Shapiro–Wilk test and Levene’s test indicate that the data distribution was normal (*p* > 0.05) and that homogeneity of variances among the groups was maintained (*p* > 0.05). The results of the repeated measures ANOVA demonstrate a significant difference between the effects of eight weeks of combined training with supplementation and combined training with placebo on changes in PON1 levels in women with Dysfunction-Associated Steatotic Liver Disease. Specifically, the mean values of this enzyme in the exercise plus supplementation group exhibited a significant increase compared to the exercise plus placebo group (8.4% vs. 1.1%) (*p* < 0.05). Similarly, a significant difference was observed regarding oxLDL levels between the studied groups, with the mean values of this index in the exercise plus supplementation group demonstrating a significant decrease compared to the exercise plus placebo group (9.6% vs. 1.3%) (*p* < 0.05) (see Table 4 and Figure 1).

The results of the statistical analysis indicate a significant difference between the effects of eight weeks of combined training with supplementation and combined training with a placebo on the lipid profile (HDL, LDL, TC, and TG) in women diagnosed with MASLD (*p* < 0.05). The mean values for LDL, TC, and TG in the exercise and supplementation group exhibited a significant decrease compared to those in the exercise and placebo group (LDL—22.5% vs. 1%; TC—13.6% vs. 2%; TG—23.3% vs. 1.1%). Conversely, the mean HDL values in the exercise and supplementation group demonstrated a significant increase relative to the exercise and placebo group (19.6% vs. 1.6%) (See Table 5 and Figure 2).

The results of the statistical analysis indicate a significant difference in the effects of eight weeks of combined training with supplementation versus combined training with placebo on AST and ALT levels in women diagnosed with Dysfunction-Associated Steatotic Liver Disease (*p* < 0.05). Specifically, the mean values of these enzymes in the exercise plus supplementation group demonstrated a notable decrease compared to the exercise plus placebo group (AST—30.9% vs. 1%; ALT—26.8% vs. 1%) (See Table 6, Figure 3).

It is noteworthy that, according to the reference values for liver enzymes (ALT 7–56 U/L, AST 10–40 U/L, CHOL < 200, HDL > 50 for females, LDL < 100—Laboratory Reference Ranges from Clinical Laboratories), alterations in the exercise group were observed alongside the consumption of royal jelly. The optimal changes were manifested within the established reference range.

It is essential to highlight that the mean values of none of the variables associated with energy intake demonstrated a statistically significant difference between the exercise plus supplement group and the training plus placebo group (Table 7). In light of the lack of substantial alterations in energy intake across macronutrient categories, it is reasonable to infer that the observed changes in the indicators of interest in this study may be attributed to the effects of combined exercise alongside the consumption of royal jelly. Consequently, the nutritional status of the subjects did not appear to influence the results obtained in this study.

## 4. Discussion

The results of the present research indicate a significant difference in PON1 levels among the subjects following the interventions (combined exercise and supplementation). The mean PON1 level in the exercise + supplement group was significantly higher than that in the exercise + placebo group, consistent with the findings of Fatolahi et al. (2017) [24], who observed six weeks of endurance and speed training in rats; Otocka-Kmiecik et al. (2021) [25], who conducted three repeated sessions of intense exercise in healthy men; Bacchetti et al. (2023) [26], who examined seven weeks of resistance training in overweight/obese individuals; and Ghasemi et al. (2017) [19], who reported on sixteen weeks of aerobic training in postmenopausal women. In contrast, these findings are inconsistent with those of Gharakhanlou et al. (2007) [27], who studied eight weeks of vigorous and moderate aerobic exercise in inactive healthy men, and Nalcakan et al. (2016) [28], who investigated aerobic exercise in trained women. The primary antioxidant enzyme carried by HDL particles is PON1 [29]. Empirical evidence suggests that HDLs protect against LDL oxidation, thereby preventing the production of pro-inflammatory oxidized lipids, primarily lipid hydroperoxides and short-chain oxidized phospholipids [30]. Given that PON1 activity is typically reduced in chronic liver diseases, including MASLD, it has been demonstrated that this reduction is associated with changes in HDL particles, the expression of peroxisome proliferator-activated receptor PPAR δ, and the regulation of monocyte chemoattractant protein-1 (MCP-1) transformation. Consequently, low levels of PON1 may be regarded as a marker of lipid peroxidation and a potential alternative indicator of increased oxidative stress and fibrosis in patients with MASLD [31]. In the present study, HDL levels increased, and oxLDL levels decreased; therefore, the current findings regarding the increase in PON1 levels are somewhat justifiable. Considering that the administration of royal jelly can mitigate oxidative stress in liver and kidney tissues, which is associated with a reduction in MDA production and an increase in the concentration of cellular antioxidant enzymes such as superoxide dismutase (SOD), catalase (CAT), glutathione reductase (GR), and glutathione peroxidase (GPx) [32], the further increase in the levels of these enzymes in the exercise + supplement group is thus accounted for.

Another significant finding was the impact of combined exercise and royal jelly supplementation on changes in LDL oxidation among women with Dysfunction-Associated Steatotic Liver Disease. This observation aligns with the results of Rezvani et al. (2017) [18], who conducted an 8-week resistance training program with postmenopausal women, Park et al. (2015) [33], who studied 12 weeks of aerobic, resistance, and traditional Korean dance training in obese older women, and Ghorbanian et al. (2021) [34], who examined 8 weeks of rope skipping combined with purslane supplementation in overweight and obese girls. Conversely, these findings do not correspond with those of Sadeghi et al. (2019) [35], who investigated the effects of eight weeks of aerobic exercise on total cholesterol (TC) and oxidized LDL (oxLDL) levels, as well as cardiovascular risk factors in obese and overweight women, and reported no significant changes in oxLDL following the exercise regimen. The discrepancies between the present study’s outcomes and those of other investigations may be attributed to factors such as the intensity, duration, and type of exercise, as well as participant characteristics. Numerous antioxidants have been shown to inhibit lipid oxidation. The principal protein in royal jelly (RJMP) exhibits several biological functions, including antibacterial, antioxidant, anticancer activities, and immune modulation. However, there is a scarcity of studies focusing on the effects of royal jelly protein hydrolysates. Some research indicates that royal jelly is rich in bioactive compounds, including 10-HDA, flavonoids (quercetin, naringenin, and galangin), phenolic acids (chlorogenic acid, caffeic acid, and ferulic acid), and essential amino acids, and it also possesses a significant inhibitory effect on oxidative DNA damage and LDL oxidation [36]. According to the researchers’ reviews, no study to date has examined the effect of royal jelly on LDL oxidation levels. Nevertheless, based on the results of the present study, it was observed that the average values of this index in the exercise plus supplement group exhibited a significant reduction compared to the placebo group. Therefore, it can be concluded that, in addition to the potential effects of exercise, royal jelly supplementation likely contributes to a reduction in LDL oxidation.

One of the most significant findings of the present study was the observed changes in the levels of low-density lipoprotein (LDL), total cholesterol (TC), and triglycerides (TG), which decreased, alongside an increase in high-density lipoprotein (HDL), in women with MASLD following 8 weeks of intervention. These results are consistent with the findings reported by Cho et al. (2014) [37], Hojjati et al. (2015) [38], Petelin et al. (2019) [39], and Barani et al. (2014) [40]; however, they diverged from the conclusions drawn by Gharakhanlou et al. (2007) [27]. Some studies indicate that engagement in physical activities, particularly aerobic exercises, and the incorporation of nutritional strategies for weight management can serve as effective interventions for improving lipid profiles [19,37]. In contrast, Gharakhanlou et al. (2007) [27] focused on the impact of aerobic exercise on Paraoxonase 1 (PON1) activity, arylesterase (ARE) activity, and serum lipoprotein profiles, reporting no significant changes in LDL-C levels or TC concentrations following aerobic interventions. The discrepancies in these findings may be attributed to variations in sample sizes, gender representation, the statistical populations studied, and the effects of the nutritional supplements utilized in the current investigation.

It is noteworthy that in the present study, combined exercises may have influenced the metabolic and physiological changes observed in muscle and liver mitochondria. Aerobic exercises significantly enhance blood flow and augment oxygen uptake by the muscles, while also improving the capacity for fat utilization in energy production and increasing the volume of oxidative enzymes within the mitochondria. Through these exercises, there is a reduction in visceral fat [41]. Resistance training promotes protein synthesis, which can enhance the volume of active muscle mass and the oxidative metabolism of muscle mitochondria. This adaptation subsequently contributes to an increase in maximum oxygen consumption and the capacity for lipolysis [42]. Aerobic and resistance exercises also lead to a reduction in pro-oxidant and pro-inflammatory proteins in liver cells. Along with these changes, oxidative enzymes that produce free radicals, such as myeloperoxidase, are decreased. This, in turn, results in an increase in PON1 expression. The increase in HDL simultaneously prevents the conversion of LDL to oxidized LDL [26]. Endurance and resistance training reduces liver inflammation and decreases hepatic fat accumulation, resulting in the diminished secretion of liver enzymes and an enhancement in liver function [43].

Royal jelly protein has been recognized as a notable factor exhibiting hypocholesterolemic properties. Royal jelly contains bile acid-binding proteins that confer a hypocholesterolemic effect. Although the influence of royal jelly on the human lipoprotein metabolism has been established, the underlying mechanisms remain poorly understood. Some research suggests that as a functional dietary supplement, royal jelly may reduce total blood cholesterol levels and, when used in a long-term regimen, improve LDL and HDL levels [40,44].

Another noteworthy outcome of this research was the changes in aspartate aminotransferase (AST) and alanine aminotransferase (ALT) levels observed in the exercise plus supplement group, which were significantly lower than those in the exercise plus placebo group. This finding aligns with the results of Kanbur et al. (2009) [45], Bahari et al. (2023) [46], and Valizadeh et al. (2011) [47], but contrasts with the findings of Barani et al. (2014) [40]. In addition to medical and pharmaceutical interventions for treating MASLD, physical activity and the use of traditional medicines with natural origins possessing anti-lipid and antioxidative properties may serve as beneficial complementary treatments, particularly in the realm of prevention. Valizadeh et al. (2011) [47] explored the effects of eight weeks of targeted aerobic training on enzyme levels (AST, ALT) in men aged 20 to 45 years with fatty liver, reporting that the training resulted in reduced AST and ALT levels in the experimental group. Additionally, Xiong et al. (2021) [48] conducted a systematic review and meta-analysis assessing the impacts of various exercise modalities on eight indices in patients diagnosed with MASLD, revealing that aerobic exercises significantly improved indices including TG, TC, LDL, HDL, ALT, AST, and body mass index (BMI). Resistance training was shown to markedly decrease AST levels, while high-intensity interval training significantly improved ALT levels in this patient population. Conversely, Barani et al. (2014) [40] examined the effects of resistance and combined training on serum liver enzyme levels and fitness indices in women with Dysfunction-Associated Steatotic Liver Disease, reporting that the level of alkaline phosphatase (ALP) significantly decreased only in the resistance training group, while no significant changes were observed in AST and ALT levels in the combined and control groups.

Studies indicate that exercise and physical activity, in conjunction with the consumption of appropriate dietary sources, effectively enhance fat metabolism, reduce blood lipids, and subsequently decrease hepatic fat accumulation [49,50]. The protective effects of royal jelly can be attributed to its antioxidant properties, which elevate the levels of hepatic antioxidant enzymes and contribute to a reduction in hepatic steatosis and liver damage [51,52]. Based on this evidence, it can be concluded that royal jelly may serve as a completely natural adjunct to physical activity, aiding in the reduction in liver enzymes and ultimately representing an effective strategy for mitigating the detrimental effects of Dysfunction-Associated Steatotic Liver Disease.

In the present study, the concurrent use of exercise and royal jelly consumption—particularly among postmenopausal women with fatty liver metabolic disease, who are at greater risk—warrants investigation. The examination of lipid–enzyme–antioxidant factors in elucidating the potential mechanisms through which exercise and royal jelly consumption exert their effects is a positive aspect of this research; however, the absence of a control group has constrained the clarity regarding the impact of each of these factors.

## 5. Conclusions

The results of this study demonstrate significant positive changes in the exercise + supplement group compared to the exercise + placebo group across all dependent variables. Consequently, it can be concluded that the combination of exercise and royal jelly supplementation may constitute an effective and recommendable strategy for mitigating the detrimental effects of Dysfunction-Associated Steatotic Liver Disease, particularly through its influence on liver enzyme activity, paraoxonase 1 levels, LDL oxidation, and lipid profile.

## Figures and Tables

**Figure 1 medicina-61-00349-f001:**
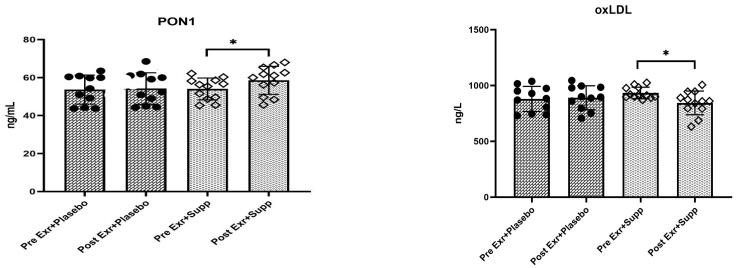
Changes of PON1 and oxLDL (means ± SD) in two groups following eight weeks of intervention. * indicates *p* < 0.05. Dot: Average individual changes in the exercise + placebo group. Squares: Average individual changes in the exercise + supplement group.

**Figure 2 medicina-61-00349-f002:**
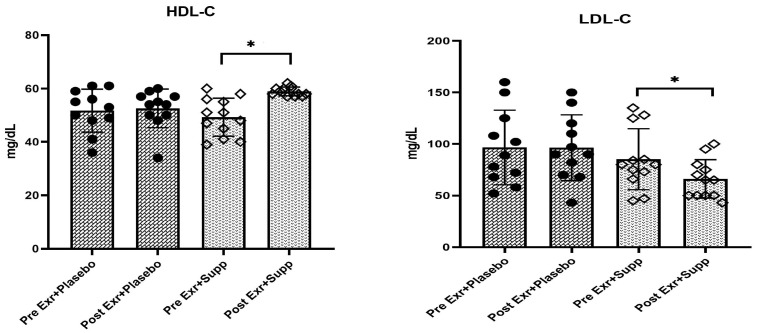
Changes in HDL, LDL, TC and TG (means ± SD) in two groups following eight weeks of intervention. * indicates *p* < 0.05. Dot/Black Squares: Average individual changes in the exercise + placebo group. White Squares: Average individual changes in the exercise + supplement group.

**Figure 3 medicina-61-00349-f003:**
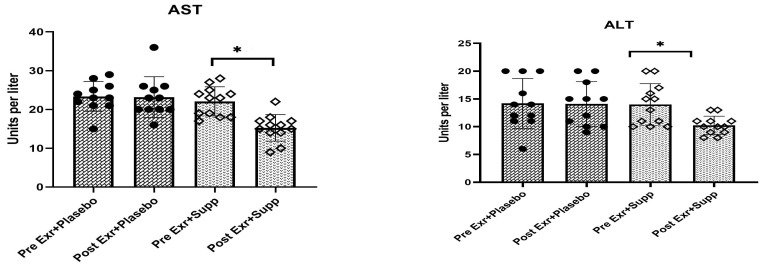
Changes of AST and ALT (means ± SD) in two groups following eight weeks of intervention. * indicates *p* < 0.05. Dot: Average individual changes in the exercise + placebo group. Squares: Average individual changes in the exercise + supplement group.

**Table 1 medicina-61-00349-t001:** Resistance training program.

Week	Repetition	Sets	Active Rest (Stretching Movements Between Each Station)	Rest Between Sets	Color of the Elastic Band	RPE
1	8–12	2	60–90 s	90 s	Yellow	10–11
2	8–12	2	60–90 s	90 s	Yellow	12–13
3	8–12	3	60–90 s	90 s	Green	12–13
4	8–12	3	60–90 s	90 s	Green	13–14
5	8–12	3	60–90 s	90 s	Blue	13–14
6	8–12	4	60–90 s	90 s	Blue	14–15
7	8–12	4	60–90 s	90 s	Red	14–15
8	8–12	4	60–90 s	90 s	Red	14–15

**Table 2 medicina-61-00349-t002:** Aerobic exercise program.

Week	1	2	3	4	5	6	7	8
Target maximum heart rate * (percentage)	70	70 s	75	75	80	80	85	85
Set and repeat	5 × 2	6 × 2	7 × 2	8 × 2	9 × 2	10 × 2	11 × 2	12 × 2
Active rest between sets (with 40% of target heart rate)	45 s	45 s	45 s	45 s	45 s	45 s	45 s	45 s

*: Target heart rate = (Reserve heart rate × training intensity percentage) + Resting heart rate. Reserve heart rate = Resting heart rate − Maximum heart rate.

**Table 3 medicina-61-00349-t003:** Demographic and physiological information of participants by group.

Variable	Group	Number	Mean ± SD
Age (years)	Exercise + Supplement	12	48.08 + 4.03
	Exercise+ Placebo	11	49.00 + 3.74
Height (cm)	Exercise + Supplement	12	162.08 + 4.66
	Exercise+ Placebo	11	158.95 + 5.04
Weight (kg)	Exercise + Supplement	12	74.92 + 10.90
	Exercise + Placebo	11	67.76 + 12.37
BMI (kg/m^2^)	Exercise + Supplement	12	28.52 + 4.10
	Exercise + Placebo	11	26.74 + 4.30
Body Fat (percentage)	Exercise + Supplement	12	20.31 + 3.78
	Exercise + Placebo	11	19.31 + 2.43

**Table 4 medicina-61-00349-t004:** Results of statistical test analysis for variables PON1 and oxLDL.

Variable	Time Effect			Time × Group Interaction Effect		
	F Value	*p* Value	Effect Size (n^2^_p_)	F Value	*p* Value	Effect Size (n^2^_p_)
**PON1 (ng/mL)**	14.24	0.001	0.404	8.71	0.008	0.293
**oxLDL (ng/L)**	3.26	0.085	0.134	5.34	0.031	0.203

**Table 5 medicina-61-00349-t005:** The results of repeated measures analysis of variance for lipid profile.

Variable	Time Effect			Time × Group Interaction Effect		
	F Value	*p* Value	Effect Size (n^2^p)	F Value	*p* Value	Effect Size (n^2^p)
**HDL (mg/dL)**	14.06	0.001	0.401	10.01	0.005	0.323
**LDL (mg/dL)**	5.91	0.024	0.22	5.69	0.027	0.213
**TC (mg/dL)**	9.95	0.005	0.321	4.54	0.045	0.178
**TG (mg/dL)**	8.7	0.008	0.293	7.33	0.013	0.259

**Table 6 medicina-61-00349-t006:** The results of repeated measures analysis of variance for AST and ALT.

Variable	Time Effect			Time × Group Interaction Effect		
	F Value	*p* Value	Effect Size (n^2^_p_)	F Value	*p* Value	Effect Size (n^2^_p_)
**AST (Units per liter)**	9.25	0.006	0.306	8.315	0.009	0.284
**ALT (Units per liter)**	7.69	0.011	0.268	6.98	0.015	0.249

**Table 7 medicina-61-00349-t007:** The results of repeated measures analysis of variance for nutritional status.

Variable	Group	Stage		Significance Level of the Time × Group Interaction	Significance Level of the Time × Group Interaction
		Pre-Test	Post-Test		
**Energy intake (kilocalories)**	Exercise + Supplement	2299.21 ± 3643.75	943.81 ± 2694.92	0.25	0.25
	Exercise + Placebo	439.90 ± 2636.73	451.96 ± 2580.10		
	*p* value	0.169	0.718		
**Fiber (Gm)**	Exercise + Supplement	17.20 ± 25.32	6.99 ± 19.78	0.823	0.823
	Exercise + Placebo	14.31 ± 25.61	13.34 ± 21.86		
	*p* value	0.966	0.64		
**Sugar (Gm)**	Exercise + Supplement	43.06 ± 87.17	25 84 ± 65.07	0.788	0.788
	Exercise + Placebo	37.97 ± 82.41	24.22 ± 65.99		
	*p* value	0.782	0.931		
**Protein (percent)**	Exercise + Supplement	2.05 ± 12.00	1.55 ± 13.25	0.339	0.339
	Exercise+ Placebo	2.06 ± 13.64	1.40 ± 13.82		
	*p* value	0.07	0.368		
**Carbohydrates (percent)**	Exercise + Supplement	12.43 ± 54.25	4.36 ± 57.67	0.212	0.212
	Exercise + Placebo	4.78 ± 62.73	5.82 ± 60.64		
	*p* value	0.046	0.178		
**Fat (percent)**	Exercise + Supplement	10.80 ± 33.58	5.26 ± 29.08	0.147	0.147
	Exercise + Placebo	5.48 ± 23.64	6.63 ± 25.09		
	*p* value	0.012	0.123		

## Data Availability

The data supporting the findings of this study are available from the corresponding author upon reasonable request.

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
