# Peer review of "Aerobic-Resistance Training with Royal Jelly Supplementation Has a Synergistic Effect on Paraoxonase 1 Changes and Liver Function in Women with MASLD"

_medicina, 2025, doi:10.3390/medicina61020349_

Round 1

Reviewer 1 Report

Comments and Suggestions for Authors

Title:  Aerobic-Resistance training with Royal Jelly supplementation has a synergistic effect on Paraoxonase 1 changes and liver function in women with NAFLD

This study highlights the possible therapeutic approach for the MASLD by aerobic Resistance training with Royal Jelly supplementation. An effective lifestyle change is recommended for a healthy life.  The study used aerobic resistance straining on the female MASH subject, and the results were promising in reducing the serum markers of MASH. Though the manuscript is well written, additional comments need to be addressed.

1.    Currently, the term non-alcoholic fatty liver disease (NAFLD) is renamed as Metabolic Dysfunction-Associated Steatotic Liver Disease (MASLD). Hence, the author must change the NAFLD to MASLD throughout the manuscript.

2.    The affiliation 3 needs to be checked for missing words.

3.    From The results, the absolute values of the analyzed parameters are given in the graph and are unclear, while the tables explain the repeated measures analysis of variance. The author must provide the absolute values of each studied parameter in a table format.

4.    It is clear from the results that the royal jelly supplementation has significantly altered the studied parameters. However, the effectiveness of aerobic resistance training is non-significant, as observed in the results of the pre and post-test of the Exercise + Placebo group. Hence, the author is expected to critically discuss it to highlight this strategy as a therapeutic lifestyle change.

5.    The author should have used another group of MAFLD subjects supplemented with royal jelly and no exercise to highlight the therapeutic applications of Aerobic-Resistance training with Royal Jelly supplementation.

6.    The entire manuscript is too wordy, and the author is expected to reduce the word count and avoid repeating information. 

Author Response

Reviewer Comments 1: I would like to express my gratitude for the insightful comments provided by the reviewer.

  1. Currently, the term non-alcoholic fatty liver disease (NAFLD) is renamed as Metabolic Dysfunction-Associated Steatotic Liver Disease (MASLD). Hence, the author must change the NAFLD to MASLD throughout the manuscript.

Response : The appropriate terminology has been substituted consistently throughout the text.

  1. Response :The affiliation 3 needs to be checked for missing words.

Response ; That has been checked ad corrected

  1. From The results, the absolute values of the analyzed parameters are given in the graph and are unclear, while the tables explain the repeated measures analysis of variance. The author must provide the absolute values of each studied parameter in a table format.

Response : All requested items for the graphs have been incorporated, and the tables have been revised accordingly.

4.It is clear from the results that the royal jelly supplementation has significantly altered the studied parameters. However, the effectiveness of aerobic resistance training is non-significant, as observed in the results of the pre and post-test of the Exercise + Placebo group. Hence, the author is expected to critically discuss it to highlight this strategy as a therapeutic lifestyle change.

Respons: Since the supplement group was not examined independently, it cannot be determined whether the observed dual effect of exercise and supplementation is attributable to the supplement, the exercise, or a combination of both. While synergistic effects were observed, financial constraints faced by the students precluded the possibility of evaluating the control group with the supplement and placebo separately. Consequently, we were only able to compare these two groups collectively.

  1. The author should have used another group of MAFLD subjects supplemented with royal jelly and no exercise to highlight the therapeutic applications of Aerobic-Resistance training with Royal Jelly supplementation.

Response: You are correct. In response to the prior comment, it was clarified that, due to financial constraints and the absence of university grant support, it was not feasible to evaluate the effectiveness of each supplement and exercise. This design should be consistently applied in future studies.

  1. The entire manuscript is too wordy, and the author is expected to reduce the word count and avoid repeating information.

Response: This matter was reviewed and addressed to the fullest extent possible.

Reviewer 2 Report

Comments and Suggestions for Authors

Remarks to the author:

In this study the authors have investigated the effects of combined aerobic-resistance training and royal jelly supplementation on liver function, lipid profile, and oxidative stress markers in postmenopausal women with non-alcoholic fatty liver disease (NAFLD). A total of 23 participants were randomly assigned to either an exercise + supplement group or an exercise + placebo group. Over 8 weeks, both groups performed resistance and aerobic exercises, with the supplement group consuming 500 mg of royal jelly on training days. Results showed significant improvements in the exercise + supplement group, including increased paraoxonase 1 (PON1) activity and HDL levels, alongside reduced oxidized LDL, total cholesterol, triglycerides, LDL, and liver enzymes (ALT, AST). According to the findings, the authors suggest that the combination of aerobic-resistance training and royal jelly supplementation may be an effective strategy for mitigating NAFLD's adverse effects.

Specific comments:

1)     Tables 3, 4, and 5: The presentation of both p and f values using a “/” needs to be checked and corrected to ensure clarity and accuracy.

2)     Figures 1, 2, and 3: The X-axes of all graphs must be labelled appropriately for better understanding and interpretation.

3)     Figure 3: The panels labelled as (a) and (b) must be reordered to follow a logical and consistent sequence.

4)     Results Section: The concluding statement, “It is important to note that the mean values of none of the variables associated with energy intake exhibited a statistically significant difference between the exercise plus supplement group and the training plus placebo group,” should be revised to clearly convey the implications of these findings.

5)     Nutritional Status: You have highlighted that “the nutritional status of the subjects did not influence the results obtained in this study.” To substantiate this claim, it would be beneficial to include the results from the nutritional status analyses, specifically those derived from the monitoring of dietary intake before and after the tests.

6)     The authors should include reference values for the parameters (eg: lipid profile, liver enzymes) analysed to clearly illustrate the extent of the differences observed between the exercise + royal jelly group and the placebo group.

7)     The authors are requested to thoroughly revise the manuscript to address the typographical errors present. 

Comments on the Quality of English Language

The authors are requested to thoroughly revise the manuscript to address the grammatical and typographical errors present. 

Author Response

Reviewer 2: I would like to express my gratitude for the insightful comments provided by the reviewer.

  • Tables 3, 4, and 5: The presentation of both p and f values using a “/” needs to be checked and corrected to ensure clarity and accuracy.

Response: The aforementioned items were organized in accordance with the esteemed judge's opinion.

  • Figures 1, 2, and 3: The X-axes of all graphs must be labelled appropriately for better understanding and interpretation.

Response: items reviewed and labeled

  • Figure 3: The panels labelled as (a) and (b) must be reordered to follow a logical and consistent sequence.

Response: Panels have been re-evaluated in accordance with the esteemed judge's recommendations.

4)     Results Section: The concluding statement, “It is important to note that the mean values of none of the variables associated with energy intake exhibited a statistically significant difference between the exercise plus supplement group and the training plus placebo group,” should be revised to clearly convey the implications of these findings.

Response:  The findings are presented in Table 6.

This sentence has been incorporated into the pertinent section. "Given the absence of substantial alterations in energy intake from macronutrient categories, it is plausible that the observed changes in the indicators of interest in this study are attributable to the effects of combined exercise in conjunction with the consumption of royal jelly.

5)     Nutritional Status: You have highlighted that “the nutritional status of the subjects did not influence the results obtained in this study.” To substantiate this claim, it would be beneficial to include the results from the nutritional status analyses, specifically those derived from the monitoring of dietary intake before and after the tests.

Response: The methodology section has been revised to incorporate detailed explanations of the assessment methods employed to evaluate the nutritional status of the participants. A statistical analysis table(6) has also been included.

6)     The authors should include reference values for the parameters (eg: lipid profile, liver enzymes) analysed to clearly illustrate the extent of the differences observed between the exercise + royal jelly group and the placebo group.

Response: Reference points derived from laboratory information were incorporated into the text, and modifications were elucidated accordingly.

7)     The authors are requested to thoroughly revise the manuscript to address the typographical errors present. 

Response: Items reviewed and revised.

Comments on the Quality of English Language

The authors are requested to thoroughly revise the manuscript to address the grammatical and typographical errors present.

Reviewer 3 Report

Comments and Suggestions for Authors

This study analyzed almost 30 patient samples to analyze whether the combination of royal jelly and exercise therapy effected on the symptoms of NASH. As a result, it was found that the combination of royal jelly and exercise therapy not only improved the symptoms of metabolic syndrome shown by down regulating the level of oxidized LDL levels, serum LDL and TG levels, but also improved serum liver injury markers such as AST and ALT.

Although this paper has some novelty, there are some issues that need to be resolved before publication.

Major comments:

1. Figures 1-3 only show the average level, making it difficult to see the variation in each sample. They should be shown as graphs that show the dots of each sample. Also, there are figures that do not show the error bars.

2. Most of the data shows no effect of exercise therapy alone (no difference between pre and post). Discussion is needed as to whether this is due to insufficient load of exercise therapy or whether there is another cause.

3. The difference such as the weight, the pathology of NASH, and age distribution of the patients are not clearly shown. It would be important to know how these were distributed in the placebo and royal jelly groups. In addition, the changes in weight in the placebo and royal jelly groups during the study should also be shown.

4. They said, It is important to note that the mean values of none of the variables associated with energy intake exhibited a statistically significant difference between the exercise plus supplement group and the training plus placebo group. Consequently, the nutritional status of the subjects did not influence the results obtained in this study.

However, no data on this issue was presented. An important point of this study is whether royal jelly improved the patient's condition by simply changing the patient's nutritional status, or whether it affected other biological responses such as immune responses, and detailed data is needed on this point.

Comments on the Quality of English Language

I have no comment about it.

Author Response

Third Honorable Reviewer
1. According to the esteemed reviewer's assessment, the figures have been revised, and the requested modifications have been implemented.
2. Although the indices assessed within the exercise group did not demonstrate statistical significance individually, the observed percentage changes have occurred in a favorable direction, which holds clinical significance. It may be worthwhile to consider the duration of the training period, as well as the intensity and volume of the exercises, as potential factors contributing to the failure to achieve the statistical significance threshold during this period. The potential reason for the lack of statistical significance of the variables observed in the combined exercise group may be attributed to the characteristics of the subjects involved in this study. Given that these individuals were patients with MASLD, the beneficial effects of exercise are likely to manifest primarily as a means of preventing deterioration and enhancing the indicators under investigation. Ultimately, these effects were reported as insignificant, which remains a crucial consideration for this patient population.
3. According to the esteemed reviewer's assessment, the requested items have been incorporated into the article as Table 3. However, given the title and objectives of the study, the interpretation of these items may be deemed unnecessary.
4. According to the esteemed reviewer's assessment, information pertaining to nutritional status has been incorporated into the article in Table 7.

Dear Editor,

I believe there has been an error concerning the third reviewer. The points raised by this reviewer appear to bear no relevance to my article, as none of them are addressed within my paper. I would appreciate your attention to this matter. Thank you for your consideration.

Round 2

Reviewer 3 Report

Comments and Suggestions for Authors

In the revised manuscript, there are several issues to be addressed for publication.

Major point

1.  I required to plot the data showing the individual points for the graphs in the review comments. However they did not changed these figures.

2. They have to add their comments about the effect of exercise therapy alone in the discussion section.

3. SD of The BMI and Body Fat ​​in Table 3 are strange (i.e. 4.10+28.52, BMI is 4.10?). Are they correct? Also, are there no significant differences in these data? If there were significant differences in these data between the two groups at the start point of treatment, this may have influenced the effectiveness of the treatment.

Author Response

  1. required to plot the data showing the individual points for the graphs in the review comments. However they did not changed these figures.
    Response: The graphs have been revised and replaced in the article, in accordance with the valuable feedback provided by the esteemed referee.
  2. They have to add their comments about the effect of exercise therapy alone in the discussion section                               Response: Regarding the potential mechanisms by which exercise influences the studied indicators, a paragraph has been included along with corresponding references. This addition is highlighted in red within the discussion section.
  3. SD of The BMI and Body Fat ​​in Table 3 are strange (i.e. 4.10+28.52, BMI is 4.10?). Are they correct? Also, are there no significant differences in these data? If there were significant differences in these data between the two groups at the start point of treatment, this may have influenced the effectiveness of the treatment.  Response: 

    Due to the accuracy of the esteemed reviewer feedback, the data were recorded in the displacement table and revised accordingly. The body mass index was assessed for two groups prior to the commencement of the study, and an independent t-test was conducted. 

    The results indicated no significant difference between the groups. Your recommendations are indeed well-founded.
